# Immune-Related Adverse Events of the Gastrointestinal System

**DOI:** 10.3390/cancers15030691

**Published:** 2023-01-23

**Authors:** Steven Nicolaides, Alex Boussioutas

**Affiliations:** 1Department of Gastroenterology, Western Health, Melbourne, VIC 3011, Australia; 2Department of Gastroenterology, The Alfred, Melbourne, VIC 3004, Australia; 3Department of Medicine, Central Clinical School, Monash University, Melbourne, VIC 3004, Australia; 4Department of Medicine, Royal Melbourne Hospital, University of Melbourne, Melbourne, VIC 3050, Australia

**Keywords:** immunotherapy, checkpoint inhibitor, colitis

## Abstract

**Simple Summary:**

Immunotherapy is an effective cancer treatment that activates the immune system to target and destroy cancer cells. The emerging use of a form of immunotherapy referred to as immune checkpoint inhibitors is often limited by the development of immune-related adverse events. This often leads to the discontinuation of treatment and limits good outcomes. In this review, we summarize the spectrum of immune-related adverse events of the gastrointestinal system with a focus on their management and emerging therapies.

**Abstract:**

Immune checkpoint inhibitors (ICI) are a form of immunotherapy that have revolutionized the treatment of a number of cancers. Specifically, they are antibodies targeted against established and emerging immune checkpoints, such as cytotoxic T-cell antigen 4 (CTLA4), programmed cell death ligand 1 (PD-L1) and programmed cell death 1 protein (PD-1) on CD8-positive T cells, which promote the destruction of tumor cells. While the immune checkpoint inhibitors are very effective in the treatment of a number of cancers, their use is limited by serious and in some cases life-threatening immune-related adverse events. While these involve many organs, one of the most prevalent serious adverse events is immune checkpoint inhibitor colitis, occurring in a significant proportion of patients treated with this therapy. In this review, we aim to broadly describe the immune-related adverse events known to occur within the gastrointestinal system and the potential role played by the intestinal microbiome.

## 1. Introduction

The development and subsequent utilization of immune checkpoint inhibitors (ICIs) have revolutionized the management of a growing number of malignancies, including metastatic melanoma, renal cell carcinoma, non-small cell lung cancer and hepatocellular carcinoma [1,2,3,4]. At the time of writing this review, the predominant ICI therapies target two pathways in immune regulation, cytotoxic T lymphocyte antigen (CTLA)-4 and programmed death (PD)-1 or its ligand (PD-L1) [5]. While this often results in a powerful anti-tumor response, it is not a targeted assault on tumor cells alone and may lead to off-target inflammation of non-malignant cells originating from multiple organ systems. The resultant pathology and impaired organ function manifests as immune-related adverse events (irAEs) commonly involving the skin, pituitary, adrenal, respiratory and gastrointestinal (GI) systems [6]. The development of irAEs has been observed in all parts of the GI system at varying rates, leading to inflammatory sequelae such as colitis, enteritis and hepatitis (Figure 1) [7]. With the growing number of indications for the use of ICI therapy, it is expected that the burden of GI irAEs is expected to continue to grow, leading to significant increases in morbidity and mortality in an already vulnerable patient population. In this review, we focus on the irAEs that are recognized to affect the gastrointestinal (GI) system.

## 2. Mechanisms of irAEs 

The immunological events that result in the development of irAEs following the commencement of immunotherapy have for the most part remained elusive. Recent and emerging evidence appears to have uncovered the origins of the immunological ‘big bang’ of the tissue resident memory (Trm) pool of the GI mucosa [8]. The Trm pool likely gives rise to the CD8+ T cells responsible, in part, for ICI colitis and possibly other luminal and barrier organ cells such as skin, lung and stomach. To a lesser extent, CD4+ T cells are thought to play a role in the pathogenesis of ICI colitis and other irAEs, along with reduced functioning of T regulatory cells [8]. There is an increase in macrophage populations which is associated with the excess production of inflammatory cytokines such as tumor necrosis factor that stimulates trafficking of immune cells, worsening the colitis [9]. It remains unclear what exactly predisposes individual patients to an increased risk of developing irAEs; however, it is likely to be a combination of genetic and environmental factors and the microbiome through T-cell recognition of mucosal bacteria that drive this susceptibility.

## 3. Immune Checkpoint Inhibitor (ICI) Colitis

The most common irAE of the gastrointestinal system is ICI colitis, with a recent systematic review highlighting an incidence of up to 40% [10,11]. The condition results in significant morbidity with symptoms such as diarrhea, abdominal pain and rectal bleeding. A systematic review conducted by Wang et al. focused on fatal toxicities of all ICI therapies using the database of the World Health Organization, reporting a total of 613 fatal events from 2009 to 2018 [6]. Out of these, 198 deaths were due to anti-CTLA-4 therapy, and ICI colitis was the cause in 135 cases (70%). With a rising incidence of cancers such as melanoma and non-small cell lung cancer where ICI therapy is already used as the standard of care, in addition to the increasing indications in other cancers, the burden of ICI colitis is expected to rise. The corollary is that there is a pressing need for effective novel therapeutic approaches that will treat the irAEs without impacting oncological outcomes. 

The main hallmark and presenting feature of ICI colitis is diarrhea [12]. Other associated symptoms depending on the presence and severity of colitis include bleeding of the rectum, abdominal pain, vomiting and pyrexia. Initial investigations include full blood count, biochemistry and assessment of stool microscopy with culture to exclude alternative causes such as infection, and endoscopy should occur early to assist with the diagnostic work-up [13].

The management is generally guided by the severity of symptoms according to the common terminology criteria for adverse events (CTCAE) (Table 1) [12]. Patients with grade 1 diarrhea or colitis are managed with supportive care and close observation. This includes advice in relation to nutrition and fluid management. It is recommended that anti-motility agents are used with caution, as they can mask disease severity and often the diarrhea is not severe enough to warrant their use [13]. ICI therapy can often be continued, and immunosuppressive therapy is not required.

For patients who develop grade 2 or greater diarrhea or colitis, management involves the commencement of corticosteroid therapy and ceasing ICI therapy. Prior to the commencement of immunosuppressive therapy, it is recommended that other causes for diarrhea are excluded, such as infection, pancreatic insufficiency and coeliac disease. It has been shown that the severity of symptoms does not correlate well with either the severity of colitis or response to treatment [14]. The combination of symptoms and endoscopic assessment including the presence of mucosal ulceration correlates better with disease severity and predicting response to treatment [15,16]. It is therefore suggested that all patients with grade 2 or higher diarrhea undergo endoscopic assessment and biopsy for histology [17].

### 3.1. The Relationship between the Microbiota, Immunotherapy and Colitis

The microbiota is a complex ecological community of commensal, symbiotic and pathogenic micro-organisms found within the gastrointestinal tract. They are thought to play an important role in both maintaining health and increasing the risk of development of certain diseases, such as clostridium difficile colitis and inflammatory bowel disease [18].

Early observations that treatment with ICI often resulted in intestinal inflammation through dysregulation of mucosal immunity prompted further research into the role of the microbiome [19]. Subsequent studies have demonstrated that the composition of the microbiota may play a role in the response to ICI therapy. There is evidence that certain bacterial signatures as well as more broadly a decrease in bacterial diversity are associated with the development of ICI colitis and that manipulation of the microbiota may lead to resolution of the colitis and improve response to ICI therapy [20]. A study of patients being treated for melanoma with ICI therapy analysis of gut microbiota found that patients with a microbiome enriched with *faecalibacterium* and *firmicutes* were more likely to develop colitis through upregulation of dendritic cells (DC) and antigen-presenting cells (APC), promoting the proliferation of T cells that enhance the blocking effect of ICI [21]. Bacteroides were protective and have been shown to hinder the blocking effect of certain ICI through stimulation of T-regulatory-cell differentiation [22]. These results are consistent with findings in other studies characterizing protective factors for the development of ICI colitis [20]. The probiotic and lactic acid bacteria lactobacillus reuteri has been shown to lead to resolution of ICI colitis, possibly through a reduction in lymphocyte distribution, modulation of cytokine production and strengthening of gut barrier function [23]. Other bacteria such as *Ruminococcaceae*, *Enterococcus faecium*, *Klebsiella pneumoniae*, *Veillonella*
*parvula* and *Akkermansia muciniphilia* have been shown to increase response to ICI therapy via multiple mechanisms of immune modulation across a number of studies [24]. The overall diversity of microbes rather than specific bacteria may impact the development of ICI colitis, with decreases in bacterial abundance being associated with other forms of autoimmunity such as hepatitis, multiple sclerosis and rheumatoid arthritis [25,26,27].

*Helicobacter Pylori* (*H. pylori*) has established itself is one of the most common chronic infections of the stomach and likely achieves this status through the establishment of an immunosuppressive environment both locally and systemically. Recent studies have demonstrated the role H. pylori infection may play on the effectiveness of ICI therapy [28]. Oster et al. found that mice not infected with *H. pylori* had better responses to ICI therapy in reducing tumor volumes following treatment compared with *H. pylori*-infected mice [29]. Analysis of retrospective human cohorts also found that *H. pylori* seropositivity was associated with reduced survival in cancer patients following treatment with ICI. As the first study to demonstrate the important role *H. pylori* infection may play in the effectiveness of ICI therapy, it raises the possibility that *H. pylori* serology or confirmation of histological diagnosis may play an important role in treatment planning and selection of therapies. It is possible that eradication of *H. pylori* therapy may influence treatment outcomes positively. The microbiome of the stomach may play a vital role in modulating the immune system that rivals the colonic bacterial population.

It is not only the bacterial organisms themselves but the presence of various metabolites that may play a role in regulating the activation of T cells. Short-chain fatty acids (SCFAs) such as butyrate and propionate have been shown to induce proliferation and differentiation of T regulatory cells, leading to higher levels of CTLA-4 and potentially increasing sensitivity to ICI therapy [30,31,32]. The use of antibiotics can significantly alter the gut microbiota and has also been shown to be associated with poor overall survival and response to treatment with immunotherapy, further highlighting that manipulation of the microbiome may influence the effectiveness of immunotherapy [33]. It is therefore feasible that manipulating the intestinal ecosystem could improve the response to immunotherapy and reduce the incidence of irAEs, and that patients showing resistance or no response may benefit from fecal microbiota transplant (FMT). The potential benefit of FMT in the treatment of refractory ICI colitis was first described in two patients. Both patients achieved complete response following either a single FMT or two FMTs [34]. The largest cohort study of 15 patients treated for refractory ICI colitis with FMT demonstrated a treatment response of 73%, and clinical remission was seen in 47% of patients [35].

### 3.2. Corticosteroids and Immunomodulators for the Management of ICI Colitis

The commencement of corticosteroid therapy should not be delayed while awaiting endoscopic assessment and is often commenced while awaiting other investigations such as stool culture to exclude infection in combination with antibiotic therapy [17]. The American Society of Clinical Oncology (ASCO) guidelines recommend that patients with grade 2 diarrhea or colitis should be commenced on oral corticosteroid therapy at a dose of 1 mg/kg/day of prednisolone, with doses of 2 mg/kg/day recommended by the national comprehensive cancer (NCCN) guidelines where oral therapy is able to be tolerated [36,37]. This is usually continued until there is improvement in symptoms, and a rapid weaning course over 4–6 weeks is commenced with the aim of minimizing the potential burden of immunosuppression to limit the negative impact to oncological outcomes. Intravenous corticosteroids such as methylprednisolone at doses of 1–2 mg/kg/day should be used in patients who are either not able to tolerate oral therapy due to nausea or vomiting or who have more severe colitis as indicated by ulceration on endoscopic assessment or ≥ grade 3 colitis. The duration of corticosteroid therapy required to see a durable response in grade 3 colitis or higher can be up to 12 weeks. There is at present no evidence or guideline recommendation for the use of pulsed corticosteroid therapy; however, the shorter duration of high-dose corticosteroid therapy may achieve similar outcomes with a reduced risk of adverse events and warrants further investigation in future studies.

### 3.3. Biologic Therapy for the Management of ICI Colitis

Up to a third of patients fail to respond to or have worsening symptoms of colitis after initiating corticosteroid therapy, with consideration given to early initiation of second-line immunosuppressive therapy [38]. Infliximab and vedolizumab are the two biologic therapies of choice, with demonstrated efficacy in the induction of remission for checkpoint inhibitor colitis [39]. Infliximab is a monoclonal antibody that binds to and neutralizes the strong pro-inflammatory effects of tumor necrosis factor (TNF). The dosing regimen has been adapted from the treatment of inflammatory bowel disease at a dose of 5 mg/kg with emerging use of higher doses of 10 mg/kg in select patients with clinical features including low levels of serum albumin or high inflammatory burden [40]. Most patients will only require one or two infusions before demonstrating a durable response, and will not require 3 or more infusions; however, if recommencement of immunotherapy is being considered, then concurrent maintenance infliximab has been described to reduce the risk of recurrence of checkpoint inhibitor colitis [41]. The addition of infliximab to steroid therapy has been demonstrated to shorten the time of symptom resolution compared to steroid therapy alone, with the added benefit of shortening the duration time of exposure to steroid therapy despite higher grades and severity of colitis in patients treated with infliximab therapy [42]. Early commencement of infliximab therapy prior to steroid failure should be considered, as supported by a large retrospective study that demonstrated a significant reduction in the failure of steroid taper and shorter courses of steroid treatment in patients who received therapy within 10 days of symptom onset [39]. The overall number of infusions in a treatment course has also been shown to influence the chances of achieving histologic remission, with patients receiving less than two infusions more likely to demonstrate evidence of colitis on subsequent biopsies of the colonic mucosa. It remains unclear whether this predicts the success of re-introduction of ICI therapy. These benefits may be overshadowed by a possible reduction in overall survival in patients treated with infliximab and steroid therapy when compared to steroid therapy alone [43].

Vedolizumab is a monoclonal antibody that binds to the GI-specific integrin α₄β₇, thereby reducing the trafficking of lymphocytes into the intestine. The largest study of the use of vedolizumab in ICI colitis demonstrated excellent response rates of 94.1% in steroid refractory patients receiving between 1–6 doses of vedolizumab at a dose of 300 mg [44].

### 3.4. Emerging and Experimental Therapies for the Management of ICI Colitis

A small number of patients (15–19%) requiring biologic therapy will be non-responders [18]. There is emerging evidence for the role of fecal microbiota transplant (FMT) in the management of refractory ICI colitis [45,46]. Differences in the composition of the gut microbiome have been demonstrated in patients with and without an anti-tumor response to ICI therapy, supporting their role in altering the immune response [47]. In rare cases of life-threatening ICI colitis and in some non-responders, the only option may be surgical management, and colectomy may be required.

The calcineurin inhibitors cyclosporin and tacrolimus inhibit T-cell activation by inhibiting the release of interleukin-2 and have proven effective in the management of inflammatory bowel disease, with cyclosporin demonstrating similar efficacy in anti-tumor necrosis factor therapy in patients with severe ulcerative colitis [48]. In a retrospective study of 11 patients who failed infliximab therapy for ICI colitis, there was a response rate of 72.7% to rescue therapy with the calcineurin inhibitors cyclosporin and tacrolimus, and three of these patients were able to recommence checkpoint inhibitor therapy [49].

Mycophenolate mofetil (MMF), which is the prodrug of mycophenolic acid, is a direct inhibitor of T and B lymphocytes. There is a single study demonstrating the use of concomitant MMF with corticosteroid therapy, with combination therapy proposed to shorten the recovery time of colitis and reduce the incidence of colitis flare following corticosteroid wean [50]. Diarrhea is a not an uncommon side effect of MMF, and this is likely to be a limiting factor in its use for the treatment of ICI colitis. 

Other therapies have been described in case reports for the successful management of refractory ICI colitis, including tofacitinib and ustekinumab; however, there are no safety data in this cohort, and their use should be considered experimental and as a last resort in those wishing to avoid surgical management [51,52,53].

### 3.5. Prevention and Recurrence of ICI Colitis in High-Risk Patients

Inflammatory bowel diseases (IBDs) such as ulcerative colitis and Crohn’s disease share similar clinical characteristics to ICI colitis. Patients with IBD can be at increased risk for the development of malignancies and may require therapy with checkpoint inhibitors; therefore, it is important to understand the unique and inherent risks posed to this patient cohort and the need to consider preventative strategies to the development of ICI colitis. In a multi-center retrospective study, Abu-Sbeih and colleagues analyzed the outcomes of 102 patients with IBD receiving ICI therapy [54]. They reported that patients with IBD were more likely to experience GI adverse events when compared to patients without underlying IBD (42% vs. 11%; *p* < 0.001), with severe (grade 3 or 4) colitis occurring in as many as one in five patients. These rates are consistent with data from a recent systematic review and meta-analysis also reporting the development of colitis in 40% of patients with IBD exposed to immune checkpoint therapy [55]. In the retrospective study by Abu-Sbeih and colleagues, it was suggested using a univariable analysis that the only risk factor for development of ICI colitis was the use of CTLA-4, and there was a trend toward IBD involvement of the colon. Further prospective data are needed to further assess the risk of disease flare in this cohort, the benefits of achieving remission of IBD prior to the initiation of immunotherapy and the most effective treatment aimed at prevention of ICI colitis.

The safe recommencement of immunotherapy following the successful treatment and remission of ICI colitis is largely guided by retrospective studies. Abu-Sbeih et al. reported 167 patients who resumed immunotherapy following ICI colitis [56]. A recurrence in ICI colitis was seen in 32% of patients who recommenced anti-PD-1/L1 therapy and in 44% of patients who recommenced anti-CTLA-4 therapy, and recurrence was typically observed earlier in onset. The risk of recurrence was higher (a) in those who experienced a longer duration of symptoms in the initial episode of ICI colitis; (b) in those whose primary episode of ICI colitis was while receiving anti-PD-1/L1 therapy and (c) in those who required second-line immunosuppressive therapy for their primary episode of ICI colitis (including the need for infliximab or vedolizumab).

Treatment strategies aimed at the prevention of ICI recurrence following the recommencement of immunotherapy has been explored in small case series. It is suggested that infliximab used in combination with immunotherapy is safe and may promote successful steroid tapering and prevent the recurrence of ICI colitis [41]. However, gut-specific immunosuppression with vedolizumab is an attractive consideration to reduce the level of systemic immunosuppression [57].

## 4. Immune Checkpoint Inhibitor Gastritis and Esophagitis

Gastritis and esophagitis caused by ICI therapy is rare, occurring in 3% of patients with upper gastrointestinal symptoms [58]. The diagnosis is made by performing esophagogastroduodenoscopy (EGD) with the combination of endoscopic and histological assessment, the exclusion of other causes for gastritis and esophagitis including *H. Pylori* infection and the use of non-steroidal anti-inflammatory medications. Once a diagnosis is made the initial treatment of choice is a proton pump inhibitor (PPI), which has been shown to lead to the complete resolution of upper gastrointestinal symptoms in 88% of patients with esophagitis [58]. In those who are PPI non-responsive, other options include corticosteroids, sucralfate or H2 receptor blockers. There is no evidence to date to demonstrate the role *H. pylori* infection may play in the development of ICI gastritis and the potential benefit of eradication therapy.

There are scarce data on the management of PPI and steroid refractory upper gastrointestinal inflammation due to immune checkpoint inhibitors. The use of infliximab and vedolizumab has been described in cases of ICI gastritis refractory to corticosteroids with complete clinical and endoscopic remission of upper gastrointestinal inflammation [59].

## 5. Immune Checkpoint Inhibitor Cholecystitis

After commencing immunotherapy, the development of right upper quadrant pain with nausea, vomiting, diarrhea and fever should raise the suspicion of immune-mediated cholecystitis. Currently, there is no way of making a definitive diagnosis without histological confirmation, and hence the treatment of cholecystitis in patients on immunotherapy should follow the traditional management of non-immune-mediated cholecystitis. Management of cholecystitis in patients on immunotherapy has been described in a case series which includes the use of antibiotics, intravenous fluids, cholecystectomy and percutaneous cholecystostomy [60].

The management of ICI-related cholecystitis with corticosteroids is associated with reduced survival, and their use is not recommended [60]. Expectant management is the preferred option, with surgery not shown to improve survival rates and should only be reserved in those patients who develop complications of cholecystitis. Immunotherapy should be recommenced following the resolution of the cholecystitis, as this is associated with longer survival [60].

## 6. Immune Checkpoint Inhibitor Pancreatic Injury (ICIPI)

Injury to the pancreas has been noted to occur in up to 4% of patients being treated with immune checkpoint inhibitors [61]. This manifests with the typical features of acute pancreatitis, including epigastric pain, nausea, vomiting, fever, diarrhea and an elevated lipase with or without imaging abnormalities. Some patients may be asymptomatic with an incidental finding of an elevated serum lipase. The management of ICIPI has limited evidence and recommendations are informed by retrospective data [61]. Corticosteroid therapy has not been shown to provide any benefit in improving short- or long-term outcomes and may increase the risk of impaired glucose tolerance in a population at risk of rapid development of pancreatic endocrine insufficiency. Admission to the hospital for a short period of supportive care with intravenous fluids and withholding ICI therapy is the most effective treatment strategy with consideration for the resumption of immunotherapy once the serum lipase has normalized, as this has been shown to improve overall survival and clinical outcomes [61].

Patients should be monitored for and managed for the development of longer-term complications, including pseudocyst formation, chronic pancreatitis, diarrhea and diabetes [62].

## 7. Immune-Checkpoint-Inhibitor-associated Coeliac Disease and Duodenitis

The development of diarrhea and abdominal pain following the commencement of ICI therapy should raise the suspicion of rare gastrointestinal irAEs in cases where colitis is excluded. Patients with positive coeliac serology (tTG-IgA) should be commenced on a gluten-free diet, which has been shown to improve symptoms and coeliac antibody titre [63]. In patients with a negative serology, EGD is recommended for endoscopic assessment and duodenal biopsies for histological diagnosis. The histology of ICI coeliac disease and duodenitis have been shown to be indistinguishable [63]. Those with coeliac disease are more likely to respond to the introduction of a gluten-free diet. In patients without response to dietary intervention, a trial of corticosteroid therapy is indicated for the management of enteritis, which is often treatment refractory and carries a poor prognosis. In refractory cases, the use of anti-TNF therapy has been described. In most cases of ICI coeliac disease, ICI therapy can be continued without interruption [63].

## 8. Immune Checkpoint Inhibitor Hepatitis

Immune checkpoint inhibitor hepatitis should be suspected in patients on immunotherapy who present with abnormal liver function tests. There are no specific biomarkers available, and it remains a clinical diagnosis that requires exclusion of other causes of liver injury, including viral hepatitis, biliary obstruction, other drug-induced liver injury and hepatic metastases.

The incidence of ICI hepatitis is reported to occur in 1–17% of patients, with rates as high as 25% when ipilimumab and nivolumab are used in combination [64]. Most patients with ICI hepatitis will present with liver function derangement within 6–14 weeks after the commencement of therapy, with a predominant elevation of the AST and ALT. Although a hepatocellular pattern is more common, cholestatic and mixed liver function test derangements can occur [65]. Generally, patients are asymptomatic but can present with non-specific symptoms such as fever and rash, or in rarer cases severe hepatitis with jaundice and liver failure [66].

The degree of hepatotoxicity is graded according to the common terminology criteria for adverse events (CTCAE) on a scale of 1–5 (Table 1), which is dependent on the degree of liver function test derangement. Similar to ICI colitis, this scoring system is used to guide management in several society guidelines [67,68].

For patients with grade 1 hepatitis, ICI therapy can be continued with close monitoring of liver enzymes with at least weekly blood tests. Corticosteroids should be commenced in patients who develop grade 2 hepatitis, with the suggested dose of 0.5–1mg/kg/day of oral prednisolone as first-line therapy. Immunotherapy should be withheld at the commencement of corticosteroid therapy with the option of recommencing therapy should the hepatitis improve to grade 1 or less.

In patients with either grade 2 hepatitis who fail to improve with oral prednisolone therapy or in grade 3 or higher hepatitis, ICI therapy should be discontinued indefinitely, hospital admission should be planned for intravenous glucocorticoid therapy and the consideration of liver biopsy should be made on a case-by-case basis. Multiple pathologic subtypes have been described in ICI hepatitis, including panlobular lymphocytic hepatitis, cholangitis, bile duct injury and fibrin ring granulomas [69]. In grade >3, hepatitis it is suggested that 1–2 mg/kg/day of methylprednisolone is administered intravenously for 48–72 h and following response a course of oral steroids weaned over 4–6 weeks. In patients who are refractory to steroid therapy, second-line immunosuppression can be considered based on expert opinion. These include azathioprine at a target dose of 1–2 mg/kg, mycophenolate 500–1000 mg twice daily in addition to steroid therapy or tacrolimus targeting a trough level of 8–10 [64]. In rare cases where the hepatitis fails to respond to second-line immunosuppression or in cases with fulminant hepatitis, then rescue therapy with anti-thymocyte globulin or trial of plasmapheresis has been described [70].

## 9. Conclusions

Currently, most of the available evidence for managing irAEs of the GI system is derived from case studies, case series and retrospective data. There is a need to support and inform both current and future management guidelines with prospective studies. In many cases, these studies are currently underway, and the results are awaited.

With the increasing use of ICI therapy, it is imperative that clinicians are able to promptly recognize the occurrence of irAEs of the GI system so that timely management can be implemented. Clinicians should be aware of the general management principles, including when to cease immunotherapy and taking the opportunity to recommence ICI therapy where it is safe to do so. The severity of colitis, which can rarely be fatal, and the use of immunosuppressive therapy must be balanced with the risks of underlying disease progression. A patient-centered approach supported by a multidisciplinary team is paramount to improving outcomes. The complexity of caring for patients with GI irAEs will necessitate the development of collaborative care teams that involves a coordinated approach between the oncologist, gastroenterologist and allied health clinicians to achieve the best outcomes in patient care.

## Figures and Tables

**Figure 1 cancers-15-00691-f001:**
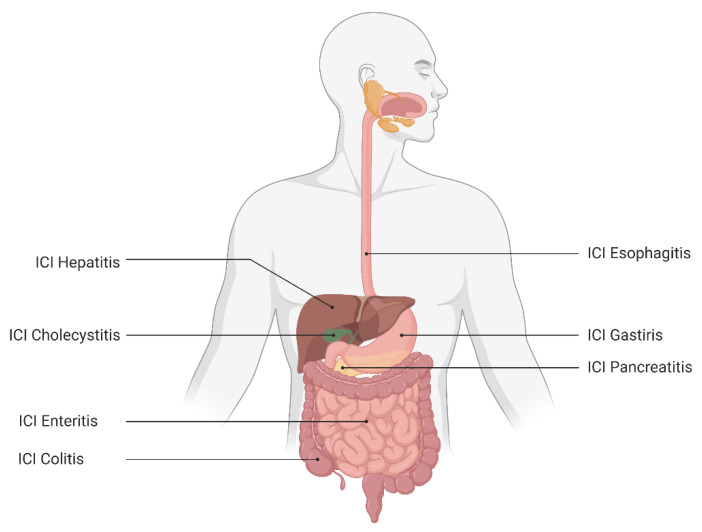
The irAEs of the gastrointestinal immune system.

**Table 1 cancers-15-00691-t001:** Common terminology criteria for adverse events (CTCAE). Definitions of irAEs of the gastrointestinal system.

CTCAE Term	Grade 1	Grade 2	Grade 3	Grade 4	Grade 5
Diarrhea	Increase of <4 stools per day over baseline; mild increase in ostomy output compared to baseline	Increase of 4–6 stools per day over baseline; moderate increase in ostomy output compared to baseline	Increase of ≥7 stools per day over baseline; hospitalization indicated; severe increase in ostomy output compared to baseline	Life-threatening consequences; urgent intervention indicated	Death
Colitis	Asymptomatic	Abdominal pain; mucus or blood in stool	Severe abdominal pain; peritoneal signs	Life-threatening consequences; urgent intervention indicated	Death
Esophagitis	Asymptomatic	Symptomatic; altered eating/swallowing; oral supplements indicated	Severely altered eating/swallowing; tube feeding; TPN or hospitalization indicated	Life-threatening consequences; urgent intervention indicated	Death
Gastritis	Asymptomatic	Symptomatic; altered GI function; medical intervention indicated	Severely altered eating or gastric function; TPN or hospitalization indicated	Life-threatening consequences; urgent intervention indicated	Death
Pancreatitis	-	Enzyme elevation; radiologic findings only	Severe pain; vomiting; medical intervention indicated	Life-threatening consequences; urgent intervention indicated	Death
Hepatitis	AST or ALT 1–2.5 × ULN and/or T-BIL 1–1.5 × ULN	AST or ALT 2.5–5 × ULN and/or T-BIL 1.5–3 × ULN	AST or ALT > 5× ULN and/or T-BIL >3 × ULN	AST or ALT > 8 × ULN	Death

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
