# Peer review of "Immune-Related Adverse Events of the Gastrointestinal System"

_cancers, 2023, doi:10.3390/cancers15030691_

Round 1

Reviewer 1 Report

The present review is well written and should add values in the scientific community. However, there are few improvements can make it more attractive: 

Few typos were observed like the improper use of commas and fullstops. Please check it carefully.

Please elaborate the simple summary section by describing the Immune checkpoint therapy and types of immune related adverse events in lay man’s term.  Then give a brief about the review.

Please reframe the sentence in line number 30-31 (At the time of writing the predominant ICI therapies target two main 30 mechanistic pathways in immune regulation including cytotoxic T lymphocyte antigen 31 (CTLA)-4 and programmed death (PD)-1 or its ligand (PD-L1).[5])

Please include reference/s for the following sentences from line number 58-63 (The main hallmark and presenting feature of ICI colitis is diarrhoea that may or may 58 not be bloody. Other associated symptoms depending on the presence and severity of 59 colitis include bleeding per rectum, abdominal pain, vomiting and pyrexia. Initial inves-60 tigations include full blood count and biochemistry and assessment of stool microscopy 61 with culture to exclude alternative causes such as infection. Endoscopy should occur early 62 to assist with the diagnostic work-up, the presence of mucosal ulceration is associated 63 with medically refractory disease.)

The authors are advised to include a section on management of IRAEs by describing methods that can be used to overcome the issue. The authors should discuss the published reports and their perspectives as well about the latest development in the field.

Author Response

Thank you to the reviewers for their insightful comments

We have subsequently revised the manuscript and provide the following reply

-The manuscript has been reviewed to check for typos and the use of comma’s and full stops.

-The simple summary section has been amended

-Sentence 30-31 has been reframed

-New references have been included

-Each section includes a discussion on the management of irAE for each part of the gastrointestinal system described in the review

Reviewer 2 Report

including (ICI) Coliti, Immune Checkpoint Inhibitor Gastritis and Esophagitis, Immune Checkpoint Inhibitor Cholecystitis, ICIPI,  Immune Checkpoint Inhibitor-associated Coeliac Disease and Duodenitis, Immune Checkpoint Inhibitor Hepatitis. Moreover, the authors made a detailed review on the above diseases from the aspects of epidemiology, mechanisms high risk factors, clinical manifestations management and emerging therapies. This review provides a good idea for further research on this topic. I have some minor comments.

minor comments:

1. I suggest that section 3 “Mechanisms of irAE and the relationship between the microbiota, immunotherapy and colitis ” is written in two sections: section A :Mechanisms of irAE; section B: the relationship between the microbiota, immunotherapy and colitis. I suggest that the content of section A goes to the introduction section or becomes a separate section.

2. Because section 3 (namely section B in comment 1) is about ICI colitis, I suggest  setting the subheadings in the following order: 2. Immune Checkpoint Inhibitor (ICI) Colitis. 2.1 The relationship between the microbiota, immunotherapy and colitis(part of section 3 of the original manuscript). 2.2. Corticosteroids and immunomodulators for the management of ICI colitis. 2.3. Biologic therapy for the management of ICI colitis. 2.4. Emerging and experimental therapies for the management of ICI colitis. 2.5. Prevention and recurrence of ICI colitis in the high risk patient.

3. Figure 1 legend should contain all the disease types in Figure 1 rather than ICI Colitis.

Author Response

Thank you to the reviewers for their insightful comments

We have subsequently revised the manuscript and provide the following reply:

-Section 3 has now been subdivided and re-ordered within the manuscript

-The legend for figure 1 has been amended